# Evaluation of Polyphenolic Composition and Antimicrobial Properties of *Sanguisorba officinalis* L. and *Sanguisorba minor* Scop.

**DOI:** 10.3390/plants11243561

**Published:** 2022-12-16

**Authors:** Alexandra-Cristina Tocai (Moţoc), Floricuta Ranga, Andrei George Teodorescu, Annamaria Pallag, Andreea Margareta Vlad, Livia Bandici, Simona Ioana Vicas

**Affiliations:** 1Doctoral School of Biomedical Science, University of Oradea, 410087 Oradea, Romania; 2Department of Food Science and Technology, Faculty of Food Science and Technology, University of Agricultural Sciences and Veterinary Medicine Cluj-Napoca, 400372 Cluj-Napoca, Romania; 3Department of Medicine, Faculty of Medicine and Pharmacy, University of Oradea, 410073 Oradea, Romania; 4Department of Electrical Engineering, University of Oradea, 410087 Oradea, Romania; 5Department of Food Engineering, Faculty of Environmental Protection, University of Oradea, 410048 Oradea, Romania

**Keywords:** *Sanguisorba officinalis* L., *Sanguisorba minor* Scop., phenolic compounds, HPLC–DAD-MS (ESI^+^), antibacterial activity, *Staphylococcus aureus*, *Escherichia coli*, *Pseudomonas aeruginosa*

## Abstract

The most widespread *Sanguisorba* species are *Sanguisorba officinalis* L. and *Sanguisorba minor* Scop. which are also found in the Romanian flora and classified as medicinal plants because of hemostatic, antibacterial, antitumor, antioxidant and antiviral activities. This study aimed to characterize and compare *Sanguisorba* species in order to highlight which species is more valuable according to phenolic profile and antimicrobial activity. Based on high-performance liquid chromatography equipped with photodiode array detection and mass spectrometry (electrospray ionization) (HPLC–DAD-MS (ESI^+^)) analysis, it was evident that the ethanol extract obtained from the leaves of *S. minor* Scop. contains the highest content of phenolic compounds at 160.96 mg/g p.s., followed by the flower and root extract (131.56 mg/g dw and 121.36 mg/g dw, respectively). While in *S. officinalis*, the highest amount of phenols was recorded in the root extract (127.06 mg/g), followed by the flower and leaves extract (102.31 mg/g and 81.09 mg/g dw, respectively). Our results show that among the two species, *S. minor* Scop. is richer in phenolic compounds compared with *the S. officinalis* L. sample. In addition, the antimicrobial potential of each plant organ of *Sanguisorba* species was investigated. The ethanol extract of *S. minor* Scop. leaves exhibited better antibacterial activity against all of the bacteria tested, especially on *Staphylococcus aureus*, with an inhibition zone of 15.33 ± 0.83 mm. Due to the chemical composition and antimicrobial effect, the *Sanguisorba* species can be used as food supplements with beneficial effects on human health.

## 1. Introduction

Medicinal plants are, therefore, important substances for the study of their traditional uses by analyzing their pharmacological effects and may be natural composite sources of new anti-infective agents [1,2,3].

The genus *Sanguisorba* belongs to the Rosaceae family and is comprised of approximately 18 to 34 species and subspecies, which are widely distributed in the northern hemisphere of Asia, North America and Europe [4]. The great burnet (*Sanguisorba officinalis* L.) and the small burnet (*Sanguisorba minor* Scop.) are both perennial plants belonging to the Rosaceae family [5,6,7]. In Romania, *S. officinalis* L. usually occurs on the wet piedmont meadows, while *S. minor* Scop. is found in more arid zones, especially dry rangelands.

*Sanguisorba officinalis (S. officinalis* L.) has been established as a herbal plant with medicinal use for a long time. It is known as Zi-Yu in Korea, Di-Yu in China, and burnet in Western countries [8,9,10,11]. However, the burnet underground organs, as well as herbs, leaves, and flowers, are used in traditional medicine. Burnet leaves are used as a decoction in the treatment of upper respiratory tract ailments, particularly pulmonary tuberculosis, which can cause bleeding. Decoction of flowers and leaves used for haemorrhoids and gastrointestinal disorders (enterocolitis, dysentery, ulcerated intestines, etc.) [12,13]. The dried root of *Sanguisorba* is listed in the Chinese Pharmacopoeia, the European Pharmacopoeia, and the Russian Pharmacopoeia in different versions [7,14,15]. The *S. officinalis* L. dried roots are used alone or mixed with other herbs for the treatment of metrorrhagia and metrostaxis, hematemesis, hematochezia, bleeding haemorrhoids, menopathy and leukorrheal diseases [16,17]. In China, *S. officinalis* L. roots are mostly used for fighting inflammation and healing skin disorders, while the whole plant is used for treating diseases in women and bloody stool haemorrhoids [17,18]. The Armenian region uses the aerial part of *S. officinalis* L. as a traditional medicine to treat various health problems, mostly digestive [18,19]. Traditional practitioners in Eastern Europe use *S. officinalis* L. to treat malignant tumours [13].

Species of *S. minor* Scop. has been used in folk medicine, in the form of infusion or tincture, for its diuretic, digestive and appetite-stimulant properties, or as a fever and diarrhea treatment. Besides its medicinal properties, the roots and aerial parts of *S. minor* Scop. are edible and are usually used in salads [7,18,20,21,22,23,24]. In Romania, *the Sanguisorba* species is also used in traditional folk medicine. *S. minor* Scop. leaves tea helps against the weakness of the digestive organs and the urinary tract, and the decoction of *S. officinalis* L. roots are used in enterocolitis and haemorrhoids [9,25]. The cultivation of burnet as a medicinal plant has a large potential. Further, *S. minor* Scop. produces high-quality forage [22,25].

Nowadays, studies have proved that *Sanguisorba* species has a variety of pharmacological activities, including anti-inflammatory [17,26,27,28], antibacterial [2,20,21,29,30,31,32], antioxidant [9,21,33,34,35,36,37,38,39], antiviral [40], anti-allergic [41], anti-cancer [42,43,44] and anti-obesity [45,46,47].

In terms of antibacterial activity, *S. officinalis* L. inhibited the activity of several bacteria, including *Bacillus subtilis* [33], *Vibrio vulnificus* [2], Methicillin-resistant *Staphylococcus aureus* [48], and *S. officinalis* L. showed significant inhibition of growth of sensitive or resistant bacteria. In contrast, *S. minor* Scop. reported significant inhibition on *Salmonella typhimurium* [49] and *Staphylococcus aureus* [21].

A very limited amount of data is available from research literature regarding phenolic compounds of the genus *Sanguisorba.* Most of the studies used the whole plant to study the composition or pharmacological properties. The novelty of this study consists of the HPLC screening of each organ (root, leaves, flowers) of both species *S. officinalis* L. and *S. minor* Scop. in order to highlight the phenolic compounds. In addition, the antimicrobial activities of each organ of *Sanguisorba* species against gram-negative bacteria (*E. coli* and *P. aeruginosa*) and Gram-positive bacteria (*S. aureus*) were evaluated.

## 2. Results and Discussion

### 2.1. Morphologic Aspects of Sanguisorba Ssp

For this study, *S. officinalis* L. and *S. minor* Scop. were collected from northwestern Romania and morphologically characterized, and the images of its plant organs are presented in Figure 1.

*S. officinalis* L. grows up to 30–150 cm (Figure 1a4), while *S. minor* Scop. grows up to a maximum of 25–55 cm tall (Figure 1b4). *S. officinalis* L. has leaves usually pinnate, with an oval shape about 5–30 cm long with 7–25 leaflets (Figure 1a2) [20,37,50]. *S. minor* Scop. has pinnate leaves, but the leaflets are in pairs placed opposite or alternative. *S. minor* Scop. have 12 to 17 pinnately compound basal leaves that are egg-shaped and sharply toothed (Figure 1b2) [38]. An inflorescence of *S. officinalis* L. consists of up to 100 flowers, each of which develops into one fruit with one seed. The inflorescence is generally terminal on elongate scapes, densely capitate or spicate, bracteates and bracteolate (Figure 1a3) [39]. The flowers are dark red, and they usually bloom from July to September [50]. At the same time, the inflorescences in *S. minor* Scop. appears at the end of stems, with a terminal spike with dense, imperfect sessile flowers (Figure 1b3). The flowers have four sepals or no petals.

Almost all *Sanguisorba* seeds are achenes, and their germination is not affected by light. The root of *Sanguisorba officinalis* L. is an irregular spindler or cylindrical and is slightly curved. The root is grayish-brown or dark brown, rough with longitudinal wrinkles, traverses cracks, and it has a thick, branched, brown rhizome (Figure 1a1) [39]. *S. officinalis radix* is often mistaken for *Polygonium cuspidatum radix* due to their similarity [37]. The radix of *S. minor* Scop. has been described as 40 cm in length with a hard, branched rhizome. (Figure 1b1) [47,49,51,52,53].

*Sanguisorba* plants are highly tolerant of congealing and droughts, allowing them to adapt easily to the environment. *S. officinalis* L. and *S. minor* Scop. are not quoted in the red lists of threatened plant species in Romania [39].

### 2.2. The Microscopic Examination of Sanguisorba Species Organs

The microscopic examination of *S. officinalis* L. and *S. minor* Scop. roots are shown in Figure 2. There are several layers of brown cells in the cork (Figure 2a1,b4). The cortex consists of several layers of oblong cells (Figure 2a1,b4). Phloem broad, with clefts (Figure 2a1). For *S. minor* Scop. the clefts are not visible. Cambium is distinct and arranged in a ring. In xylem, vessels are arranged radially, surrounded by fibers (Figure 2a1,b4). Parenchymatous cells contain abundant starch granules scattered with clusters of calcium oxalate (Figure 2a1,b4). A thin layer of striated cuticle covers the epidermis. The xylem occupies a large portion of the stem and is four to five times wider than the phloem. It is composed of vessels, fibers and xylem parenchyma (Figure 2a2,b5). There are one-cell-wide medullary rays that run radially from the cambium through the xylem to the pith. Trichomes are mostly absent for both species. The diagrammatic section of the leaf shows that the sheathing base is a crescent shape in outline (Figure 2a3,b6). In cross-section, the upper (adaxial) and lower (abaxial) epidermis of the leaf is one-layered and covered externally by a thin layer of the cuticle (Figure 2a3,b6). Between the epidermis, ground tissue is made up of parenchyma cells with chlorophyll pigments. In the mesophyll, there are 1–2 layers of compressed palisade cells, followed by 8 layers of spongy parenchyma cells with large air spaces between them (Figure 2a3,b6). Xylem consists of vessels, tracheids, and parenchyma cells, while phloem is made up of sieve cells and parenchyma cells. The leaf of *S. minor* Scop. is covered on both sides with non-glandular trichomes (Figure 2b6), more densely on the abaxial side. The non-glandular trichomes are conical with a pointed tip and slightly base, straight curved. For *S. officinalis* L. leaf the non-glandular trichomes are not present.

### 2.3. Screening of Phenolic Compounds Using HPLC–DAD-MS (ESI^+^)

Twenty-three compounds belonging to the classes of tannins, phenolic acids and flavonoids were identified in the various organs of each Sanguisorba species (Table 1). The HPLC chromatograms of roots, leaves and flowers from *S. officinalis* L. and *S. minor* Scop. were shown in Appendix A, respectively.

The most representative phenolic compounds in *Sanguisorba* ssp.’organs are represented in Figure 3.

Among the two species studied, *S. minor* Scop. contains 1.5 times more tannins than *S. officinalis* L. The flowers and roots are the organs rich in tannins in both species of *Sanguisorba*, compared to the leaves. The compounds Galoyl-bis-hexahydroxydiphenyl–glucoside, were not detected in the leaves and flowers of *S. officinalis* L. and *S. minor* Scop. In the roots of *S. officinalis* L. (SOR), 2, 3-Hexahydroxydiphenoyl-glucose dominates (37.13% of total roots tannins), while in the roots of *S. minor* Scop. (SMR), Punicalagin gallate dominates (69.05%). Both species have Punicalagin gallate in their leaves, although it is more abundant in SMF leaves than SOF leaves, while Sanguiin H-1 is the predominant tannin in the flowers in proportions of 47.52% and 62.89% for *S. officinalis* L. (SOF) and *S. minor* Scop. (SMF), respectively. Punicalagin is an ellagitannin with the largest molecular weight and most abundant polyphenol known to be responsible for antioxidant activity [54]. Sanguiins, as one of the subgroups of polyphenolic ellagitannins, exhibit various pharmacological activities due to having different chemical structures. These compounds possess a broad spectrum of pharmacological features such as antibacterial, antifungal, antiviral, anti-inflammatory, antioxidant and osteoprotective [8] were highlighted for Sanguiin compounds.

Ten compounds from the class of flavonoids derivate from the subclass of flavanols and flavonols, and two belonging to the class of anthocyanins were identified. Only flavonols were identified in the root, of which the dominant was B-type (epi) catechin dimer isomers of 47.926 mg/g dw and 50.632 mg/g dw in SOR and SMR, respectively. A polyphenol compound, catechins have been extensively investigated and proven to be antioxidants by scavenging free radicals and retarding extracellular matrix degradation caused by ultraviolet radiation and pollution. [55]. In our study, catechin is the predominant flavonoid present in the root of *S. officinalis* L (SMR) in proportions of 34.50%.

Alternately, the leaves and flowers contain both flavanols and flavonols, the predominant being quercetin-glucosides in SOL and SOF, while quercetin-glucuronide is the predominant compound in SML. Gatto et al., [56] found out that *S. minor* Scop. leaves are rich sources of quercetin-3-glucoside and kaempferol-3-glucoside which comprise 52% of total phenolic compounds using the HPLC method. In addition, apigenin derivatives and chlorogenic, caffeic and chicoric acid derivatives were also detected.

Among *Sanguisorba* spp. organs, caffeic acid glucoside was identified as the dominant phenolic acid, in descending order in SMR, SOF, and SOR of 46.68%, 42.03%, and 33.36% of total phenolic acids, respectively. Caffeic acid has been greatly employed as an alternative strategy to combat microbial pathogenesis and chronic infection induced by microbes such as bacteria, fungi, and viruses. Several studies have demonstrated the effectiveness of caffeine acid in combating microbial pathogenesis and chronic infection caused by bacteria, fungi, and viruses [57].

Ellagic acid was identified only in SOR and SMR roots in lower amounts compared to other phenolic acids, but ellagic acid hexoside was identified as predominant in the leaves with SML (34.82% of total leaves tannins) and SOL (23.87% of total leaves tannins). It has been demonstrated that ellagic acid derivatives are capable of inhibiting a wide range of microbial pathogens as well as anti-biofilm properties. A variety of microbial pathogens, such as: *P. aeruginosa*, *E. coli* and methicillin-resistant *Staphylococcus aureus* were inhibited by ellagic acid derivates [58].

Also, p-Coumaroylquinic acid is found in greater quantity in SMF (53.47%), and SOL (28.79%). According to Biernasiuk et al., there is a difference between the composition of phenolic compounds in rhizomes and leaves of *S. officinalis* L., who identified free and bounded phenolic acids (gallic and ellagic acids) using 2D-TLC method [59].

Ayoub [60] also identified 2-(4-carboxy3-methoxystyryl)-2-methoxysuccinic acid, quercetin, ellagic acid and kaempferol in ethanolic extracts of the whole plant of *S. minor* Scop. using NMR and ESI-MS spectral analysis.

As a result of HPLC analysis, SOR were found to contain higher total phenols than SMR (128.81 mg/g dw and 122.07 mg/g dw, respectively), which is similar to the results obtained in our previous study [9], where Folin-Ciocalteu was used to measure total phenols content.

Planting region, growing environment and harvest season affect the phenols content in *S. officinalis* L. and *S. minor* Scop. sample. *S. officinalis* L. was collected from Săcădat commune, a grassland area, while *S. minor* Scop. was collected from the village of Bucea, which is characterized by a hill and meadow relief. The moisture content of a hill will be higher, and as the saturation level comes faster, the rains will be much more abundant, whereas grasslands have significant variations in temperatures with hot summers and cold winters. Thereby, precipitation is moderate. It is important to consider climatic factors, as they can affect biochemical biosynthesis and, therefore, bioactive compound concentration. Also, the variations of temperature, water stress or radiation prior to and/or during the harvest could justify the differences found between *Sanguisorba* spp.

SML and SMF contain the highest levels of total phenols, indicating that *S. minor* Scop. is a rich source of bioactive compounds.

### 2.4. Antimicrobial Activity

The screening of some antibiotics against gram-positive and gram-negative bacteria (Appendix A) was used to identify which is the most effective in terms of antimicrobial activity. Among the antibiotics tested, Ciprofloxacin had the strongest antimicrobial activity against *E. coli*, while Meropenem was effective against *P. aeruginosa* and Cefoxitin and Clindamycin were effective against gram-positive bacteria (*S. aureus*) (see Appendix A).

The antimicrobial potential of extracts of each plant organ (roots, leaves and flowers) of both *Sanguisorba* ssp. was evaluated, and the results are shown in Table 2.

It can be noted that all *Sanguisorba* spp. extracts exhibited varying degrees of antibacterial activity against all bacterial strains tested. Among the organs of the *Sanguisorba* spp., the highest antimicrobial activity against *S. aureus* in descending order were SML > SOL > SMF. Higher antimicrobial activity was found in SMF, SOL, and SMR samples against *E. coli*, while only the SOL sample showed high diameter inhibition against *P. aeruginosa*. No significant differences were recorded for antimicrobial activity against all bacteria investigated in the case of the SOR sample. Instead, in the case of SMR, a higher antimicrobial activity against *E. coli* was observed compared with others bacteria. The highest diameter of inhibition was recorded in the case of leaves of *S. minor* Scop. sample against *S. aureus*, while the leaves of *S. officinalis* L. exhibited against *E coli*. and *P. aeruginosa*. According to HPLC analysis of SM leaves, total phenolic content is higher than that of SO (186.56 mg/g dw and 101.66 mg/g dw, respectively), indicating high antimicrobial effectiveness.

An effective antimicrobial activity was also obtained in the case of flowers of *S. minor* Scop. (SMF) against *E. coli* and *S. aureus* compared to SOF sample. SMF was characterized by a greater amount of Sanguin H1 compared to SOF, a compound with antimicrobial activity [8].

Cirovic et al., 2022, [21] investigated the antimicrobial potential of methanol and chloroform extracts of *S. minor*. All tested strains (*B. cereus, E. faecalis, S. aureus, E. coli, P. aeruginosa, E. aerogenes, P. mirabilis, K. pneumoniae, S. enteridis* except *C. albicans*, were more susceptible to the methanol extract than the chloroform extract. Our results are in agreement with the findings of Cirovic et al., 2022, who reported good antimicrobial activity of *S. minor* Scop. aerial parts against Gram-positive bacteria. The most sensitive was *S. aureus.* However, in this particular case, it was evaluated only *S. minor* Scop. aerial parts and it wasn’t compared to *S. officinalis* L. [21].

In another report, Karkanis et al., 2019, observed a higher antimicrobial capacity of *S. minor* Scop. roots extract on *Bacillus cereus*, *Enterobacter cloacae*, *Escherichia coli*, *Listeria monocytogenes*, *Staphylococcus aureus*, and *Salmonella typhimurium* in comparison to extracts from the leaves and stems [22]. In contrast to the results of the present investigation, Finimundy et al., 2020, reported as well, a higher antibacterial effect of *S. minor* Scop. roots extracts on *S. aureus*, *B. cereus*, *L. monocytogenes*, and *S. typhimurium* in comparison with the leaves [49]. Do et al., 2005, using the disc diffusion method for testing the antibacterial activity of *S. officinalis* L. extract, and their results showed relatively strong antimicrobial activities against *B. subtilis*, *S. typhimurium*, *P. aeruginosa*, *S. aureus* (more than 15 mm inhibition zone) [32]. Chen et al., 2015, found that the ethanol extract of *S. officinalis* L. inhibited the biofilm formation ability of an MRSA (Methicillin-resistant *Staphylococcus aureus*) strain which indicates that with the combined use of the right antibiotics and the extract of *S. officinalis* L., the dose of antibiotic used in the treatment could be minimized [48]. Ginovyan et al., 2017, used an agar well diffusion assay for the evaluation of the antimicrobial properties of plant materials. Acetone and methanol extracts of *S. officinalis* L. aerial part inhibited the growth of the tested strains (*E. coli*, *P. aeruginosa*, *S. aureus*, *S. typhimurium*, *Candida albicans*), whereas crude extracts of the root did not show activity against *Salmonella* at tested concentration [31]. Zhu et al., 2020, revealed that purified polyphenolic extract from *S. officinalis* L. determined antibacterial activity against *B. subtilis* [61]. Although studies have shown that *S. officinalis* L. has strong antimicrobial activity [31,62] in comparison with *S. minor* Scop., in our case, *S. minor* Scop. has better antimicrobial activities, especially the leaves, than *S. officinalis* L. extracts. *S. minor* Scop. leaves contained the greatest amount of phenolic compounds, primarily punicalagin gallate, quercetin-glucuronide, and ellagic acid hexoside, which have antimicrobial properties. Punicalagin and ellagic acid have been demonstrated by Venusova et al., 2021 to have antimicrobial activity against *S. aureus*, *P. aeruginosa* and *E. coli* [63], which are in agreement with our results because ellagic acid was present in a high concentration in leaves of *S. minor* Scop.

The antibacterial activity of *S. minor* Scop. leaves extract could be related to their phenolic components, particularly quercetin-glucuronide, as well as *S. officinalis* L. leaves extract but rich in quercetin-glucoside, which from literature data it was demonstrated that quercetin derivates could be capable of inhibiting the growth of several bacteria [56].

*Sanguisorba* is considered a ‘Herb that stops bleeding’ in Traditional Chinese Medicine. The name *Sanguisorba*, where “sangui” is related to “sanguine”, which means “blood red”, and ‘‘sorba’’ means “to staunch”, these plants stop haemorrhages and echymosis by having hemostatic properties [9,64,65]. In traditional medicine, different anatomical parts of *S. officinalis* L. and *S. minor* Scop. has been used to treat various symptoms, some of which are presented in Table 3.

The biological effects of plants of the genus *Sanguisorba* are mainly due to their phytochemical composition. Data from the literature suggest that these plants are rich sources of triterpenoids and phenolic compounds.

*S. officinalis* L. and *S. minor* Scop. were used in traditional medicine for various affections, characterized in particular by bleeding or inflammations. These traditional uses can be related to the amount of tannins found in these species because tannins are considered valuable plant secondary metabolites for which they are used in the versatile fields of treatment [18,20]. Tannins have potential antioxidant, antimicrobial, antiviral, antimutagenic, antihelmintic, and hepatoprotective effects [69]. Punicalagin gallate and Sanguiin H-1 are the main tannins in *Sanguisorba* spp., contributing to anti-inflammatory, antimicrobial, antiviral, and antioxidant properties [18,20,70]. The effectiveness of medicinal plants rich in tannins in treating various ailments can be explained either by their properties or by their synergistic effects with other bioactive polyphenols.

An overview of the chemical constituents of *Sanguisorba* spp. and their biological effects selected from the literature are presented in Table 4.

There are a few compounds in the literature and in our plants that are common in terms of antibacterial activity: ellagic acid, caffeic acid and catechin.

According to the other authors, caffeic acid was found in the roots of *S. minor* Scop., ellagic acid in the aerial parts of *S. minor* Scop. and catechin in the roots of *S. officinalis* L. [18,20,22,35,37,60]. In our case, catechin was identified plenty in the roots of *S. officinalis* L., while ellagic acid and caffeic acid in *S. minor* Scop. roots.

## 3. Materials and Methods

### 3.1. Plant Material

The selected medicinal plants were collected from Bihor and Cluj country, Romania. *S. officinalis* L. August 2021 from Săcădat village, Bihor country situated between 47°03′23″ N, 22°16′48″ E, and *S. minor* Scop. was collected in May 2021 from Bucea village, Cluj country situated between 46°96′30″ N, 22°68′05″ E. Săcădat village is up to 163 m in altitude and the relief is plain, whereas Bucea village is up to 648 m in altitude and the relief is hilly and meadow. Since *S. officinalis* L. flowers in August, the harvest was done in that month, while *S. minor* Scop. flowers more quickly from May to June, after which it loses its flowers and seeds, leaving only the leaves. Because we examined the roots, leaves, and flowers of each species in our study, we considered the flowering period for *Sanguisorba* spp.

The identification of *S. officinalis* L. and *S. minor* Scop. plants were made at the Department of Pharmaceutical Botany at the University of Oradea, Faculty of Medicine and Pharmacy. A specimen of each *Sanguisorba* ssp. was kept in the Herbarium of the Faculty of Medicine and Pharmacy Oradea, Romania, registered in NYBG Steere Herbarium, under the code: Uop 05 367-*S. minor* Scop. and Uop 05 368-*S. officinalis* L.

### 3.2. Macroscopic and Microscopic Examination of Sanguisorba officinalis L. and Sanguisorba minor Scop.

The macroscopic analysis between *S. officinalis* L. and *S. minor* Scop. was based on the determination of morphological characters and aimed at the stability of their identity.

From the collected roots, stems and leaves of *Sanguisorba officinalis* L. and *Sanguisorba minor* Scop. samples were made into microscopical sections. Microscopic analysis of the samples was conducted using the OPTIKA B-383PL light microscope (SC Nitech SRL, Bucuresti, Romania), equipped with Proview digital camera and software.

Cross sections were made at the level of fresh roots, stems and leaves (10×) according to the standard methods.

### 3.3. Determination of Phenolic Profile by HPLC–DAD-MS (ESI^+^) 

Standards of Chlorogenic Acid, Ellagic acid, Rutin, Catechin and Cyanidin were purchased from Sigma-Aldrich (United States of America). Acetonitril (HPLC Grade) was purchased from Merck (Germany) and the Ultra-Pure Water Was Purified with the Direct-Q UV from Millipore (U.S.A). The Other Chemical Reagents Used in the Experiment Were Analytical Grade

#### 3.3.1. Phenolic Compounds Extraction for Chromatographic Analysis

1 g of the crushed sample was extracted with 10 mL of 70% ethanol by vortexing 1 min at Heidolph Reax top vortex, followed by 30 min ultrasonic treatment. The extract was stored in the refrigerator at 4 °C for 24 h and then was centrifuged at 10,000 rpm for 10 min and T = 240 °C at Eppendorf AG 5804 centrifuge. The above operations were repeated 3 times, and the extract was accumulated and concentrated on a Heidolph low pressure rotary evaporator to a total volume of 10 mL.

The supernatant was filtered through a 0.45 μm Chromafil Xtra nylon filter and 20 μL was injected into the HPLC system.

#### 3.3.2. HPLC–DAD-MS (ESI^+^)—Analytical Conditions

Agilent 1200 HPLC system equipped with quaternary pump, solvent degasser, autosampler, UV-Vis photodiode detector (DAD) coupled with Agilent single quadrupole mass detector (MS) model 6110 (Agilent Techologies, Santa Clara, CA, USA) was used.

The positive ionization mode was applied to detect the phenolic compounds; different fragmentor, in the range 50–100 V, were applied. The column was a Kinetex XB-C18 (5 μm; 4.5 × 150 mm i.d.) from Phenomenex, USA.

The mobile phase was (A) water acidified by formic acid 0.1% and (B) acetonitrile acidified by formic acid 0.1%. The following multistep linear gradient was applied: start with 5% B for 2 min; from 5% to 90% of B in 20 min, hold for 4 min at 90% B, then 6 min to arrive at 5% B. Total time of analysis was 30 min, flow rate 0.5 mL/min and oven temperature 25 ± 0.5 °C.

Positively charged ions were detected using the Scan mode of mass spectrometry. The applied experimental conditions were: gas temperature 350 °C, nitrogen flow 7 L min, nebulizer pressure 35 psi, capillary voltage 3000 V, fragmentor 100 V and *m*/*z* 120–1200.

Chromatograms were recorded at wavelength λ = 280 nm, λ = 350 nm and data acquisition was done with the Agilent ChemStation software (B.02.01SR2, Santa Clara, CA, USA).

In order to identify the phenolic compounds, retention time, UV–Vis absorption and mass spectra were compared with those of the standard compounds and with data from the literature for phenolic compounds [36,76,77]. As a result of the spectral characteristics of phenolic compounds, the wavelength λ = 280 nm is distinctive to some phenolic acids, flavanol monomers and polymers, while wavelength λ = 320 nm to hydroxycinnamic acids and flavonols [77]. In order to quantitate phenolic compounds, a calibration curve was constructed using standard compounds (gallic acid, chlorogenic acid, and rutin) at concentrations ranging from 1 to 100 μg/mL. The regression coefficients of calibration curves ranged between 0.9937 and 0.9981.

### 3.4. Preparation of Extracts for Antimicrobial Test

The plant parts, SOR (*S. officinalis* L. root), SOL (*S. officinalis* L. leaves), SOF (*S. officinalis* L. flowers), *SMR (S. minor* Scop. root), SML (*S. minor* Scop. leaves), SMF (*S. minor* Scop. flowers) were washed with distilled water to get rid of any unwanted debris and dust. For 7 days, dust-free parts were dried in the shade until they were dry enough to grind. An electrical grinder was used to grind each part of the plants separately into an even powder. Crude extract was prepared using the Soxhlet method. Each powder of samples (10 g) were kept in a reciprocating shaker for 72 h for continuous mixing at a speed of 200 rpm. During this study, ethanol (70%) was used as an organic solvent for extraction. In the end, the crude extracts were filtered with muslin cloth and Whatman no. 1 filter paper and then concentrated using a vacuum rotary evaporator. Dried extracts were dissolved in distillated water and stored at −20 °C in sterile containers.

#### 3.4.1. Bacterial Strains and Disc Diffusion Method

The following standard strains of Gram-positive and Gram negative bacteria were used in this study: *S. aureus* (ATCC, Manassas, VA, USA, 25923), *E. coli* (ATCC, Manassas, VA, USA, 25922), and *P. aeruginosa* (ATCC, Manassas, VA, USA, 10662). The strains used in this study were prepared in 0.9% sterile saline and adjusted as an inoculum to a final concentration of 0.5 McFarland standard. Twenty milliliters of Mueller-Hinton agar were homogenized with 20 µL of microbial suspension and then poured into a Petri dish. The pots were left at room temperature for 15 min to allow the culture medium to solidify [78].

A variation of the Kirby-Bauer diffusion method, a simple and rapid technique that determines the spectrum of sensitivity/resistance to antibiotics of microorganisms, was used for quantificative evaluation of antimicrobial activity, a variant in which the free discs were loaded with 50 µL of the extracts [79]. The test was applied for all extracts obtained from plant organs of *S. officinalis* L. and *S. minor* Scop. sample.

To prepare the inoculum, 3–5 colonies from a culture plate were homogenized in sterile physiological serum after 18 h, in the stationary growth phase, to obtain standard turbidity. To control the density of the inoculum, we used a 0.5 McFarland standard (optical density at 550 nm is 0.125) as well as a Densimat digital densitometer. The suspensions were homogenized by stirring a vortex for 15–20 s. The suspension was calibrated by adding different amounts of isotonic chlorinated solution; once the optical density was achieved, 1/10 solution was made. Using a Drigalski rod, 500 µL of the 1/10 dilution of the 0.5 McFarland suspension was dispersed on top of the agarized Muller-Hinton medium. As soon as the plates were sowered, they were left for 3–5 min to absorb the inoculum before applying SOR, SOL, SOF, SMR, SML, and SMF extracts on a 50 µL space on disk [79]. After the spots were placed and the volume was placed in each well, the plates were thermostated at 35 ± 2 °C, under aerobic conditions, for 16–18 h.

In order to read and interpret the results, we used a graduated ruler to measure the diameter of the inhibition zones in the reflected light, on the back of the Petri dish.

#### 3.4.2. Statistically Analysis

The samples of each organ (roots, leaves and flowers) of *S. officinalis* L. and *S. minor* Scop. were analyzed, and all assays were performed in triplicate. The data of analysis are represented as mean value ± standard deviation (SD).The data were subjected to analysis by one-way ANOVA (Tukey’s multiple comparison test) at *p* < 0.05 significant level.

## 4. Conclusions

Both species of *Sanguisorba* spp. are rich sources of bioactive compounds from the class of tannins, flavonoids and phenolic acids. Following the analysis of the polyphenols profile, certain characteristic features of these plants can be distinguished. *S. minor* Scop. contains 1.5 times more tannins than *S. officinalis* L., the roots and leaves being the organs richest in these compounds. The leaves and flowers contain both flavanols and flavonols, the predominant being quercetin-glucosides in the leaves and flowers of *S. officinalis* L., while quercetin-glucuronide is the predominant compound in the leaves of *S. minor* Scop. that have been certified as having antibacterial activity. The antimicrobial assay showed the remarkable antimicrobial potential of the *S. minor* Scop., highlighting its efficacy in inhibiting *S. aureus* bacterial strains. Although in the specialized literature, studies related to *S. officinalis* L. predominate, the results of this study demonstrated that *S. minor* Scop. also represents a rich source of bioactive compounds, especially tannins with strong antimicrobial potential. Further studies are needed to find out the mechanism by which these plants exert their antimicrobial effect and to identify the compounds responsible for this effect. Because a variety of secondary metabolites are present in plants, a possible synergistic action between the compounds could be possible for the exercise of different biological functions.

## Figures and Tables

**Figure 1 plants-11-03561-f001:**
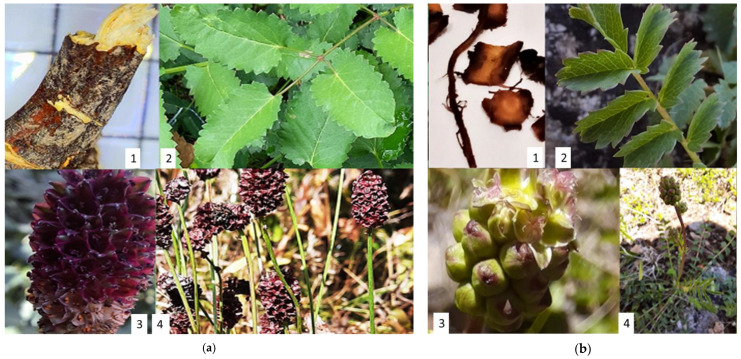
(**a**). *S. officinalis* L. from Bihor county, RO, where 1-Root, 2-Leaves, 3-Flower, 4-Stems and flowers. (**b**). *S. minor* Scop. from Cluj county, RO where 1-Root, 2-Leaves, 3-Flower, 4-Stems, flowers, leaves (personal photos).

**Figure 2 plants-11-03561-f002:**
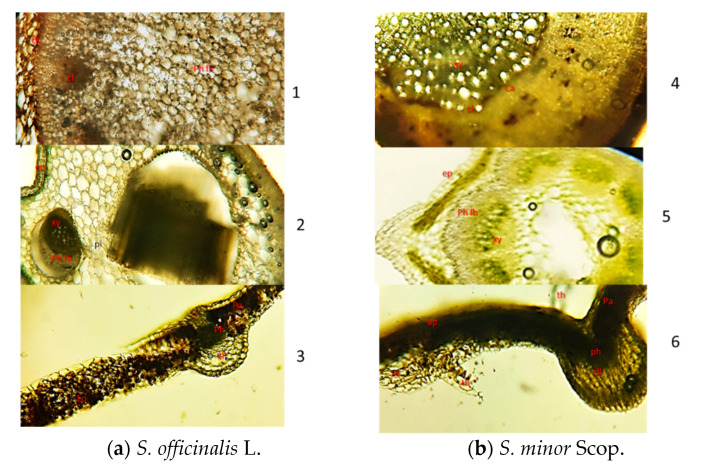
(**a**). Transverse section of different organs of *S. officinalis* L. (**a1**) root (SOR); (**a2**) stem (SOS); (**a3**). leaf (SOL); (**b**) Transverse section of different organs of *S. minor* Scop. (**b4**) root (SMR); (**b5**) stem (SMS); (**b6**) leaf (SML); Ck—cork, Cf—clefts; Co-cortex, Ph fb-phloem fibres, Ph—phloem, Ca—cambium, Cl—cluster of calcium oxalate, Xy—xylem; Ep—epidermis, Pi—pith; Pa—palisade cells; Up—upper epidermis; Lo—lower epidermis; Th—trichome; Cll—collenchyma; Cr—crystal shealth; SOR—*S. officinalis* L. root; SOS—*S. officinalis* L. stems; SOL—*S. officinalis* L. leaf; SMR—*S. minor* Scop. root; SMS—*S. minor* Scop. stem; SML—*S. minor* Scop. leaf.

**Figure 3 plants-11-03561-f003:**
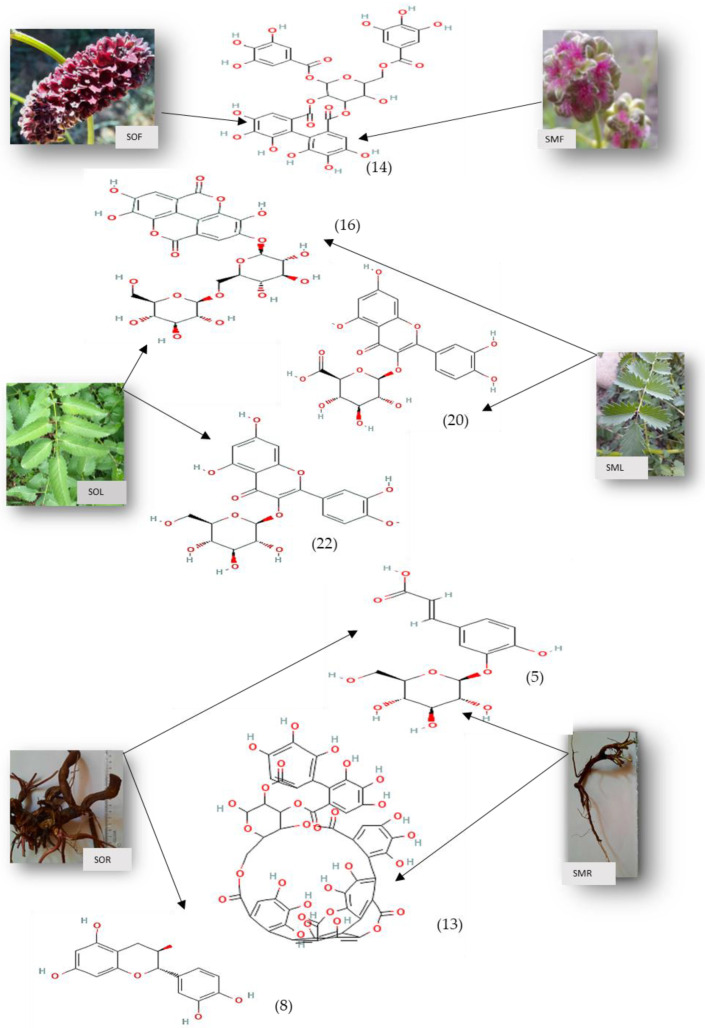
The chemical structure of predominant phenols isolated from organs of *S. officinalis* L. and *S. minor* Scop. (**14**)—Sanguiin H-1; (**16**)—Ellagic acid hexoside; (**20**)—Quercetin-glucuronide; (**22**)—Quercetin-glucoside; (**5**)—Caffeic acid-glucoside; (**13**)—Punicagalacin gallate; (**8**)—Catechin https://pubchem.ncbi.nlm.nih.gov/ (accessed on 29 September 2022).

**Table 1 plants-11-03561-t001:** HPLC–DAD-MS (ESI^+^) profile of phenolic compounds (mg/g dw) in roots, leaves and flowers of *S. officinalis* L. and *S. minor* Scop.

No. Peak	Compounds	RT (min)	UV λmax (nm)	[M+H]^+^ (*m*/*z*)	SOR	SOL	SOF	SMR	SML	SMF
							Tannins			
1	2,3-Hexahydroxydiphenoyl-glucose	3.05	272	483, 303	15.3 ± 1.27 ^bc^	6.95 ± 0.13 ^f^	12.64 ± 0.37 ^de^	15.12 ± 0.21 ^c^	11.59 ± 0.38 ^e^	17.82 ± 0.6 ^a^
12	Sanguiin H-10 derivative	13.84	235	1265, 303	9.01 ± 0.58 ^a^	1.95 ± 0.025 ^e^	8.29 ± 0.045 ^b^	5.00 ± 0.06 ^d^	1.75 ± 0.036 ^ef^	6.76 ± 0.02 ^c^
13	Punicalagin gallate	14.24	236	1253, 303	3.76 ± 0.025 ^f^	9.23 ± 0.161 ^e^	11.13 ± 0.087 ^d^	28.58 ± 0.047 ^a^	25.93 ± 0.02 ^b^	18.67 ± 0.02 ^c^
14	Sanguiin H-1	14.58	234	786, 303	7.69 ± 0.043 ^ef^	8.27 ± 0.015 ^e^	19.67 ± 0.035 ^b^	9.53 ± 0.02 ^d^	13.46 ± 0.045 ^c^	26.03 ± 0.056 ^a^
15	Galoyl-bis-hexahydroxydiphenyl –glucoside, isomer 1	15.01	268	935, 303	2.73 ± 0.032 ^b^	nd	nd	5.11 ± 0.22 ^a^	nd	nd
16	Ellagic acid hexoside	15.12	357	465, 303	nd	2.28 ± 0.036 ^bc^	3.37 ± 0.02 ^b^	nd	11.49 ± 0.043 ^a^	0.89 ± 0.036 ^d^
17	Galloyl-bis-hexahydroxydiphenyl-glucoside, isomer 2	15.33	268	935, 303	2.83 ± 0.02 ^b^	nd	nd	4.28 ± 0.036 ^a^	nd	nd
19	Ellagic acid pentoside	15.62	357	435, 303	0.29 ± 0.041 ^d^	nd	2.60 ± 0.264 ^a^	0.24 ± 0.025 ^de^	1.16 ± 0.026 ^b^	1.02 ± 0.058 ^bc^
			Flavonoids							
2	C-type (epi)catechin trimer	9.96	279	867, 291	10.48 ± 0.045 ^a^	nd	nd	6.27 ± 0.02 ^b^	nd	nd
4	Cyanidin-glucoside	11.17	520	449, 287	nd	nd	0.49 ± 0.04 ^a^	nd	nd	0.13 ± 0.07 ^b^
7	B-type (epi)catechin dimer, isomer 1	13.53	276	579, 291	17.23 ± 0.01 ^a^	6.08 ± 0.04 ^d^	5.06 ± 0.037 ^e^	3.44 ± 0.035 ^f^	13.38 ± 0.03 ^b^	10.26 ± 0.025 ^c^
8	Catechin	12.57	280	291	20.40 ± 0.045 ^a^	7.43 ± 0.01 ^d^	2.90 ± 0.035 ^f^	8.58 ± 0.02 ^c^	15.42 ± 0.026 ^b^	6.42 ± 0.015 ^e^
9	Cyanidin-malonylglucoside	13.12	517	535, 287	nd	nd	0.25 ± 0.037 ^a^	nd	nd	0.06 ± 0.026 ^b^
11	B-type (epi)catechin dimer, isomer 2	11.98	276	579, 291	11.19 ± 0.041 ^ab^	2.04 ± 0.035 ^f^	5.76 ± 0.026 ^d^	11.79 ± 0.05 ^a^	9.77 ± 0.037 ^c^	3.71 ± 0.037 ^e^
18	Quercetin-galloyl-glucoside	15.46	350	617, 303	nd	2.89 ± 0.03 ^b^	6.21 ± 0.135 ^a^	nd	1.95 ± 0.025 ^c^	1.08 ± 0.015 ^cd^
20	Quercetin-glucuronide	16.32	360	465, 303	nd	nd	nd	nd	20.20 ± 0.026 ^a^	8.33 ± 0.02 ^b^
22	Quercetin-glucoside	16.10	360	479, 303	nd	18.62 ± 0.051 ^a^	9.26 ± 0.032 ^b^	nd	8.17 ± 0.092 ^cd^	8.61 ± 0.045 ^c^
23	Kaempferol-glucuronide	17.14	346	463, 287	nd	7.62 ± 0.015 ^a^	1.88 ± 0.035 ^cd^	nd	6.26 ± 0.028 ^b^	2.12 ± 0.096 ^c^
			Phenolic acids							
3	3-Caffeoylquinic acid (Neochlorogenic acid)	10.45	322	355	3.06 ± 0.08 ^bc^	1.00 ± 0.105 ^f^	2.08 ± 0.068 ^de^	3.42 ± 0.02 ^ab^	3.79 ± 0.04 ^a^	2.11 ± 0.02 ^cd^
5	Caffeic acid-glucoside	11.31	323	343	8.68 ± 0.04 ^b^	2.35 ± 0.04 ^e^	5.36 ± 0.228 ^c^	10.90 ± 0.02 ^a^	4.73 ± 0.015 ^d^	2.01 ± 0.026 ^ef^
6	5-Caffeoylquinic acid (Chlorogenic acid)	11.43	322	355	4.64 ± 0.025 ^a^	1.57 ± 0.03 ^f^	2.24 ± 0.045 ^d^	2.10 ± 0.032 ^de^	3.73 ± 0.01 ^bc^	3.96 ± 0.055 ^b^
10	p-Coumaroylquinic acid	13.36	323	339, 165	7.34 ± 0.025 ^c^	2.75 ± 0.03 ^ef^	3.07 ± 0.043 ^e^	4.08 ± 0.07 ^d^	8.09 ± 0.075 ^b^	11.48 ± 0.036 ^a^
21	Ellagic acid	16.20	360	303	2.30 ± 0.026 ^ab^	nd	nd	2.85 ± 0.03 ^a^	nd	nd

Where, SOR—*S. officinalis* L. root; SOL—*S. officinalis* L. leaves; SOF—*S. officinalis* L. flowers; SMR—*S. minor* Scop. root; SML—*S. minor Scop*. leaves, SMF—*S. minor* Scop. flowers. Data are expressed as the mean value ± SD (*n* = 3). Different letters superscripts in the same line indicate significant differences between the samples (*p* < 0.05). nd-not detected.

**Table 2 plants-11-03561-t002:** Antimicrobial activity of tested extracts from the roots, leaves and flowers of *S. officinalis* L. and *S. minor* Scop.

	Bacteria Strains	*S. aureus*	*E. coli*	*P. aeruginosa*
Samples				
Diameter of inhibition zone (mm) *				
SOR		5.50 ±0.81 ^d, A^	6.56 ± 1.2 ^d, A^	6.50 ± 0.75 ^b, A^
SOL		8.43 ± 0.75 ^b,B^	11.46 ± 0.73 ^ab, A^	11.33 ± 1.15 ^a, A^
SOF		2.50 ± 0.91 ^e, B^	2.66 ± 0.75 ^e, B^	4.60 ± 0.88 ^bc, A^
SMR		1.26 ± 0.77 ^ef, A^	10.53 ± 0.75 ^bc, B^	1.06 ± 0.47 ^ef, A^
SML		15.53 ± 0.83 ^a, A^	1.06 ± 0.68 ^ef, B^	2.00 ± 0.36 ^de, B^
SMF		8.40 ± 1.04 ^bc, B^	11.56 ± 0.9 ^a, A^	3.70 ± 1.11 ^cd, C^

Where, SOR-*S. officinalis* L. root; SOL-*S. officinalis* L. leaves; SOF-*S. officinalis* L. flowers; SMR-*S. minor* Scop. root; SML-*S. minor* Scop. leaves, SMF-*S. minor* Scop. flowers.* Data are expressed as the mean value ±SD (*n* = 3). Lowercase letters mean significant differences (*p* < 0.05) on the column (differences between plant organs and the same microorganism), while the different uppercase letters mean significant differences ((*p* < 0.05) on the line (between the same organ compared to the three investigated bacteria).

**Table 3 plants-11-03561-t003:** Traditional use of different parts of *S. officinalis* and *S. minor*.

Scientific Name	Common Name	Plant Part Used	Traditional Uses	References
*Sanguisorba officinalis* L.	Great Burnet	Aerial part	Fevers and bleeding.	[13,20,28,66]
		Underground parts	Treatment of hematochezia, bleeding haemorrhoids, bloody flow, burns, sores and skin ulcers	[18,66]
		Whole plant	Ulcerative colitis, diarrhea, dysentery, and bladder problems	[13,18]
*Sanguisorba minor* Scop.	Small Burnet	Aerial herb	Sunburn, eczema, seasoning for salads	[20,67,68]
		Underground parts	Commonly used for treating diarrhea, dysentery and has protective effect against stomach ulcers and fungi with high blood sugar lowering effects.	[18,20]

**Table 4 plants-11-03561-t004:** Polyphenols and their biological effects from literature identified in *Sanguisorba* spp.

Compounds	*Sanguisorba* spp.	Biological Effects	References
	Flavonoids		
(−) epicatechin	*S. officinalis* L.	antioxidant	[18,68]
Apigenin derivates	*S. minor* Scop.	antihyperglycemic, anti-inflammatory	[18,22]
Quercetin	*S. officinalis L. and S. minor* Scop.	antioxidant, anti-inflammatory	[18,20,71]
Quercetin-3-glucuronide	*S. officinalis* L. *and S. minor* Scop.	antioxidant, anti-inflammatory	[7,72]
(+)-gallocatechin	*S. officinalis* L.	antioxidant, antihyperglycemic	[18,20,71,73]
(+)-Catechin	*S. officinalis* L.	antibacterial, antioxidant	[20,37]
Kaempferol	*S. officinalis* L.	antioxidant, antineoplastic	[20,71]
Isorhamnetin	*S. officinalis* L.	anti-inflammatory, antioxidant, antitumor	[18,73]
Taxifolin	*S. officinalis* L.	antibacterial, anti-inflammatory	[18,73]
Cyanidin 3-glucoside	*S. officinalis* L.	antioxidant, anti-inflammatory, anticarcinogenic	[20,71]
	Phenolic acids		
Chlorogenic acid	*S. minor* Scop.	antidiabetic effect, neuroprotective effect	[18,22]
Caffeic acid	*S. minor* Scop.	antibacterial, anti-inflammatory antoxidant,	[18,20,22,35]
Ellagic acid	*S. minor* Scop.	antibacterial, antioxidant, anti-inflammatory, antiproliferative properties	[20,60]
Gallic acid	*S. officinalis* L.	antibacterial, antifungal, antiviral activities, antioxidant, anti-inflammatory, and antineoplastic properties	[20,64]
	Tannins		
Sanguiin H-1	*S. officinalis* L.	antioxidant	[20,74,75]
Sanguiin H-2	*S. officinalis* L.	antioxidant	[20,74,75]
Sanguiin H-6	*S. officinalis* L.	antibacterial, antioxidant and anti-inflammatory	[20,75,76]
Sanguiin H-10	*S. officinalis* L.	antioxidant	[20,22,74]

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
