# Peer review of "Evaluation of Polyphenolic Composition and Antimicrobial Properties of Sanguisorba officinalis L. and Sanguisorba minor Scop."

_plants, 2022, doi:10.3390/plants11243561_

Round 1

Reviewer 1 Report

The presented study demonstrates interesting data on the bioactive potential of Sanguisorba plants, some minor revision is necessary. In tables 1 and 2 please indicate below the full names of acronyms of plant extracts (SOR, SOL etc). Authors should identify more clearly the significance of this study for bioactive compound usages in traditional medicine and connect the part on traditional use more efficiently with their data.

Author Response

The authors would like to thank Reviewer 1 for taking the time to provide us with useful suggestions that improve the quality of the manuscript.

Our response to the comments are included in the attached document.

Thank you very much!

Reviewer 2 Report

This article is devoted to the phytochemical and antimicrobial study of two species of Sanguisorba. The article is of interest to specialists in the field of phytochemistry and biomedical chemistry of natural substances. The topic fits the requirements of the journal, the amount of experimental data is sufficient for publication. However, I have some suggestions for improving the content of this manuscript:

1. It is not clear from the methodology why the authors used 70% ethanol. Is this the method developed by the authors? Has the extraction parameters been varied? If the authors use a known technique, then it is necessary to refer to it. In the literature, when extracting various components from herbaceous plants, completely different solvents and their ratios are used.

2. The authors were able to investigate the antimicrobial effect of the obtained extracts. This is an important result. However, it is necessary to specify in more detail by what specific substance a similar level of antimicrobial activity is achieved. This may be important for further studies in the preparation of antimicrobial substances.

3. Conclusions can be extended to give them a more fundamental look.

4. What is the main achievement of this work?

5. Please cite: 10.3390/molecules27186129.

Author Response

The authors would like to thank Reviewer 2 for taking the time to provide us with useful suggestions that improve the quality of the manuscript.
Our response to the comments are included in the attached document.
Thank you very much!
